# Testis single-cell RNA-seq reveals the dynamics of de novo gene transcription and germline mutational bias in *Drosophila*

Evan Witt, Sigi Benjamin, Nicolas Svetec, Li Zhao*

Laboratory of Evolutionary Genetics and Genomics, The Rockefeller University, New York, United States

**Abstract** The testis is a peculiar tissue in many respects. It shows patterns of rapid gene evolution and provides a hotspot for the origination of genetic novelties such as de novo genes, duplications and mutations. To investigate the expression patterns of genetic novelties across cell types, we performed single-cell RNA-sequencing of adult *Drosophila* testis. We found that new genes were expressed in various cell types, the patterns of which may be influenced by their mode of origination. In particular, lineage-specific de novo genes are commonly expressed in early spermatocytes, while young duplicated genes are often bimodally expressed. Analysis of germline substitutions suggests that spermatogenesis is a highly reparative process, with the mutational load of germ cells decreasing as spermatogenesis progresses. By elucidating the distribution of genetic novelties across spermatogenesis, this study provides a deeper understanding of how the testis maintains its core reproductive function while being a hotbed of evolutionary innovation.
DOI: https://doi.org/10.7554/eLife.47138.001

## Introduction

The testis is a highly transcriptionally active tissue whose core function of sperm production is conserved across kingdoms. In humans, flies, and mice, spermatogenesis consists of several key steps: (1) differentiation of germline stem cells into spermatogonia, (2) mitotic divisions of spermatogonia, which become spermatocytes, (3) meiotic divisions to generate primary spermatids, and (4) sperm maturation (*Fuller, 1993*; *Jan et al., 2012*; *White-Cooper, 2010*). Across animal species, the testis is unique from a transcriptomics perspective because it expresses more genes than any other tissue (*Parisi et al., 2004*). Genotypes and phenotypes associated with sex and reproduction diverge rapidly and may have important functional consequences (*Lande, 1981*). Despite evolutionary genetic hypotheses trying to explain the complexity of the testis transcriptome, it remains unclear why this tissue expresses a broader array of genes than any other tissue, including the brain, which is more phenotypically and structurally complex and contains more cell types (*Parisi et al., 2004*; *Soumillon et al., 2013*; *Parisi et al., 2003*).

Not only it is a highly transcriptionally active tissue, the testis is also a hotspot for newly originated genes (*Long et al., 2003*; *Neme and Tautz, 2016*). One hypothesis is that testis catalyzes the birth and retention of novel genes (*Kaessmann, 2010*). This hypothesis suggests that novel genes are likely to be born in testis due to a permissive chromatin state. Novel functional genes with beneficial products are selectively preserved and eventually evolve more refined regulatory programs (*Bai et al., 2007*; *Kaessmann, 2010*). In the past decade, many studies have found that young genes, including de novo originated genes (genes born from ancestrally noncoding DNA), tend to be biased towards the testis (*Levine et al., 2006*; *Long et al., 2003*; *Brown et al., 2014*; *Ruiz-*

*For correspondence:
lzhao@rockefeller.edu

**Competing interests:** The authors declare that no competing interests exist.

*Orera et al., 2015*; *Tautz and Domazet-Lošo, 2011*; *Zhao et al., 2014*). De novo genes may arise in two main ways: (1) an unexpressed DNA sequence gains expression, and the resulting transcript can acquire a function, coding potential or (2) a potential Open Reading Frame (ORF) gains expression and translation, and undergoes functional refinement (*Carvunis et al., 2012*; *Durand et al., 2019*; *McLysaght and Hurst, 2016*; *Schlötterer, 2015*). Natural selection may not only preserve testis-bias and function of novel genes, but also shape expression and function in somatic tissues for others (*Chen et al., 2010*; *Chen et al., 2013*; *Zhou et al., 2008*). Elucidating the biology of new-gene evolution therefore requires a comprehensive picture of spatio-temporal dynamics of testis gene regulation.

Spermatogenesis is a highly conserved process in many animal taxa and is well-understood from an anatomical and histological perspective, but its molecular foundations are still poorly understood (*Birkhead et al., 2008*; *Demarco et al., 2014*; *Russell et al., 1993*; *White-Cooper, 2010*). New analytical methods in genomics allow the quantification of expression biases of gene groups involved in various cellular processes (*Jung et al., 2018*; *Lukassen et al., 2018*; *Stévant et al., 2018*). From the prevalence of their transcripts, one can make inferences about the developmental timing of translation, DNA repair, nuclear export, and other processes. Moreover, these methods also make possible the identification of germline variants and the individual cells in which they occur.

Such methods include the recent advent of single-cell sequencing, a technology that may shed light on unknown aspects of germline mutation. For instance, it is known that the human mutation rate per base per generation ranges from 10E-7 to 10E-9 (*Moorjani et al., 2016*; *Scally and Durbin, 2012*), but this germline mutation rate is the result of an equilibrium between errors/lesions and repair. Substitutions that arise within an individual's germline but do not reach mature gametes will not be passed to the next generation, meaning that population genetics approaches can only observe a subset of germline variants. Is the population-level mutation rate influenced by the mutation-repair equilibrium of spermatogenesis?

A roadblock to the answer of this question is the fact that any substitutions that prevent gamete maturation or fertilization will be lost from the population, meaning that the population-level mutation rate may vastly underestimate the germline variants propagating within individuals. Since male *Drosophila* do not undergo meiotic recombination, germ cell variants that occur in earlier developmental stages may not be repaired through recombination related mechanisms (*Hunter, 2015*). It is also known that different cell types in the testis accumulate DNA lesions at different rates (*Gao et al., 2014*), but it is unclear if the net mutational load varies during spermatogenesis. Single-cell RNA-seq can be used to infer mutational events within a whole tissue, even if such lesions would be repaired before gamete maturation. Unlike single-cell genome sequencing, this approach can infer the cell types associated with each variant, allowing estimation of the mutational load of cells as they progress through spermatogenesis. Due to its versatility, reproducibility, and wealth of useful data, single-cell RNA-seq is a powerful tool for the study of germline mutation.

We leveraged single-cell RNA-seq and unsupervised clustering to identify all the major cell classes of the sperm lineage, validated by previously studied marker genes. We identified populations of somatic cells, including cyst stem cells, hub cells, and terminal epithelial cells. We found that the overall gene expression is very active in early spermatogenesis and decreases throughout spermatogenesis. Lineage-specific de novo genes (genes derived from ancestrally noncoding DNA [*Zhao et al., 2014*]) showed expression in various cell types and are commonly expressed in spermatocytes. We also identified putative germline de novo substitutions from our population of cells and found that they decrease in relative abundance during spermatogenesis. We also found that the proportion of mutated cells decreases throughout spermatogenesis, a finding with possible implications for the study of male germline DNA repair. In an opposite pattern, DNA damage response genes are upregulated in early spermatogenesis, indicating a role for these genes in early spermatogenesis.

These patterns of mutation and de novo gene expression augment and enrich our current understanding of the male-specific evolutionary novelty. It was previously known that young de novo genes tend to be testis biased, and we have further traced the main source of this bias to spermatocytes. We uncover a compelling time course of mutational load throughout spermatogenesis, putting forward the *Drosophila* testis as a model for the study of spermatogenic mutational surveillance. Mutation and de novo gene evolution are critical components of the adaptive process, and our results demonstrate these processes in action during spermatogenesis.

## Results

### Unsupervised clustering elucidates the distribution of de novo genes across cell types

We prepared a single-cell suspension from freshly dissected testes of 48-hours-old *D. melanogaster* adult males (*Figure 1—figure supplement 1*, also see Materials and methods). The cell suspension was then made into a library and sequenced. We recovered 426,563,073 reads from a total of 5000 cells. On average, we mapped 85,312 reads per cell and detected the expression of an average of 4185 genes per cell. The dataset correlates well with bulk testis RNA-seq and a separate testis single-cell RNA-seq library, with a Pearson's R of 0.97, indicating high reproducibility (*Figure 1—figure supplement 2*). Using t-Stochastic-Neighbor Embedding (t-SNE) in Seurat (*Van Der Maaten and Hinton, 2008*; *Satija et al., 2015*) we reduced the dimensionality of the gene/cell expression matrix to two primary axes and grouped cells by their similarity across their thousands of unique gene expression profiles. Grouping similar cells into clusters, we observed marker genes enriched in particular clusters, allowing us to infer the identity of the cells within each cluster (see Materials and methods).

Based on the clustering results, we inferred the presence of germline stem cells, spermatogonia, spermatocytes, and spermatids (germ cells) as well as cyst stem cells, terminal epithelial cells, and hub cells (somatic cells) (*Figure 1A and B*). We confirmed that the top 50 most highly enriched genes in cell clusters from each cell type (*Supplementary file 1*) were consistent with previous knowledge of marker genes. For instance, *cup* genes were biased toward late spermatids (*Barreau et al., 2008*), and *Hsp23* and *MtnA* were highly expressed in the epithelial cells (*Faisal et al., 2014*; *Michaud et al., 1997*). Cell clusters from each developmental stage in the t-SNE map are near each other, suggesting that cell progression through spermatogenesis is a continuous process. The expression of marker genes confirmed the assignment of cell clusters (*Figure 1C and D*). Germline Stem Cells (GSCs) and early spermatogonia clustered together due to 1) high transcriptional similarity, 2) the relatively low numbers of GSCs within the tissue, and 3) the sparse expression of GSC-specific marker genes. Different types of somatic cells clustered close to each other in the t-SNE graph, suggesting distinct transcriptional patterns compared to germ cells. A principal component analysis of variable genes in the testis is presented in *Figure 1—figure supplement 3*.

To gauge the accuracy of our cell type assignments, we queried if various cell types utilize biological pathways known to be important in spermatogenesis. Using a PANTHER Gene Ontology (GO) search of all significantly enriched genes for each cell type, we found that the most enriched GO terms for GSC, early and late spermatogonia tend to involve translation, transcription, and ATP synthesis (*Supplementary file 2*), supporting high levels of cellular activity. Early spermatocytes showed an enrichment for ubiquitin-independent proteasomal catabolism; late spermatocytes were enriched for genes involved in spermatid motility and differentiation (*Supplementary file 2*). Early spermatids were enriched in GO terms for spermatogenesis, gamete generation, and cellular movement, and late spermatids showed no enrichment in any GO terms (*Supplementary file 2*).

The average number of expressed genes per cell ranged between ~2000 genes for late spermatids and ~7000 genes for our late spermatogonia (*Figure 2A*). The number of genes expressed in early and late spermatids is lower than at any other point during the sperm lineage (*Figure 2A*), suggesting that post-meiotic transcription exists, but occurs at a lower level (*Schultz et al., 2003*). Consistently with this result, the cellular RNA content, measured by the number of Unique Molecular Indices (UMI) recovered per cell, is low in spermatids and high in spermatogonia and early spermatocyte (*Figure 2B*). The RNA content in the post-meiotic cells is five times lower (21%) than that of meiotic stages, inferred from the average number of UMIs per cell. Congruently, we noticed that spermatids express 53% of the total number of genes that spermatocytes express.

Since most de novo genes in *Drosophila* are expressed in the testis (*Zhao et al., 2014*), we asked whether they can be found uniformly across cell types, or whether they are enriched in particular stages of spermatogenesis. We detected expression of 87 segregating and 97 fixed de novo genes from *Zhao et al. (2014)* that are expected to have originated sometime after the divergence with *D. simulans* (*Zhao et al., 2014* identified 142 segregating and 106 fixed genes, respectively). Consistent with our predicted expression patterns of functional novel genes, we found that de novo genes are expressed in various cell types and a large number of de novo genes are expressed in meiotic germ cells (*Figure 2C*).

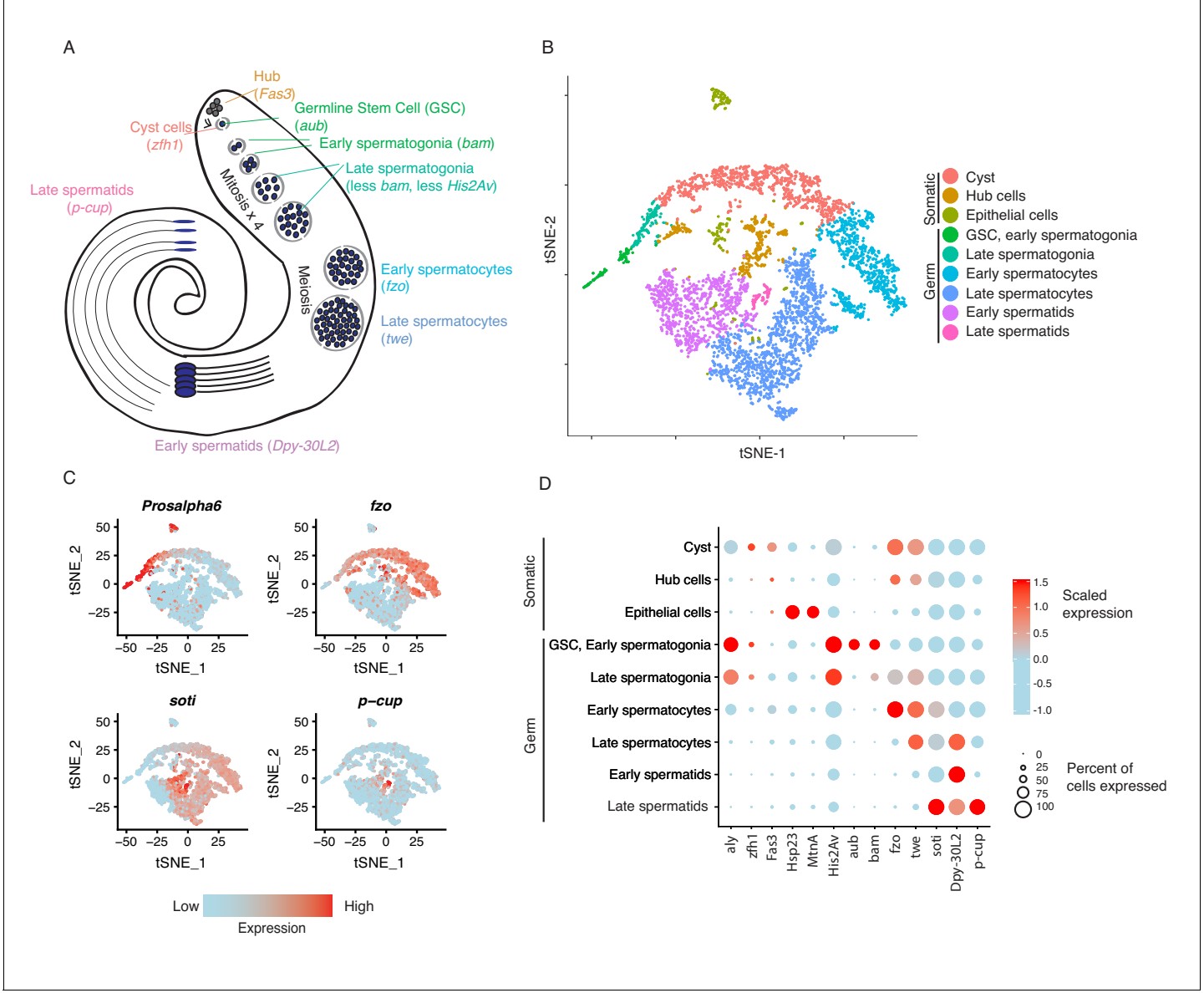

**Figure 1.** Clustering and cell-type assignment of single cells in Seurat. (**A**) An illustration of the major cell types in the testis, and the marker genes we used to identify them are in brackets. Somatic cells are hub, cyst, and epithelial cells. Spermatogenesis begins with germline stem cells which undergo mitotic divisions to form spermatogonia. These become spermatocytes which undergo meiosis and differentiate into spermatids. (**B**) A t-SNE projection of every cell type identified in the data. (**C**) Examples of marker genes that vary throughout spermatogenesis. *His2Av* is most active in early spermatogenesis, *fzo* and *soti* are active in intermediate and late stages, respectively, and *p-cup* is exclusively enriched in late spermatids. (**D**) Dotplot of scaled expression of marker genes in each inferred cell type. The size of each dot refers to the proportion of cells expressing a gene, and the color of each dot represents the calculated scaled expression value; blue is lowest, red is highest. 0 is the gene's mean scaled expression across all cells and the numbers in the scale are z scores. The cutoffs shown here were chosen to emphasize cell-type-specific enrichment of key marker genes. The genes used to assign each cell type are detailed in the Materials and methods section.

DOI: https://doi.org/10.7554/eLife.47138.002

The following figure supplements are available for figure 1:

**Figure supplement 1.** Establishing a single cell suspension from *Drosophila* testes.

DOI: https://doi.org/10.7554/eLife.47138.003

**Figure supplement 2.** Reproducibility of RAL517 single-cell sequencing data.

DOI: https://doi.org/10.7554/eLife.47138.004

**Figure supplement 3.** Principal component analysis of testis-expressed genes.

DOI: https://doi.org/10.7554/eLife.47138.005

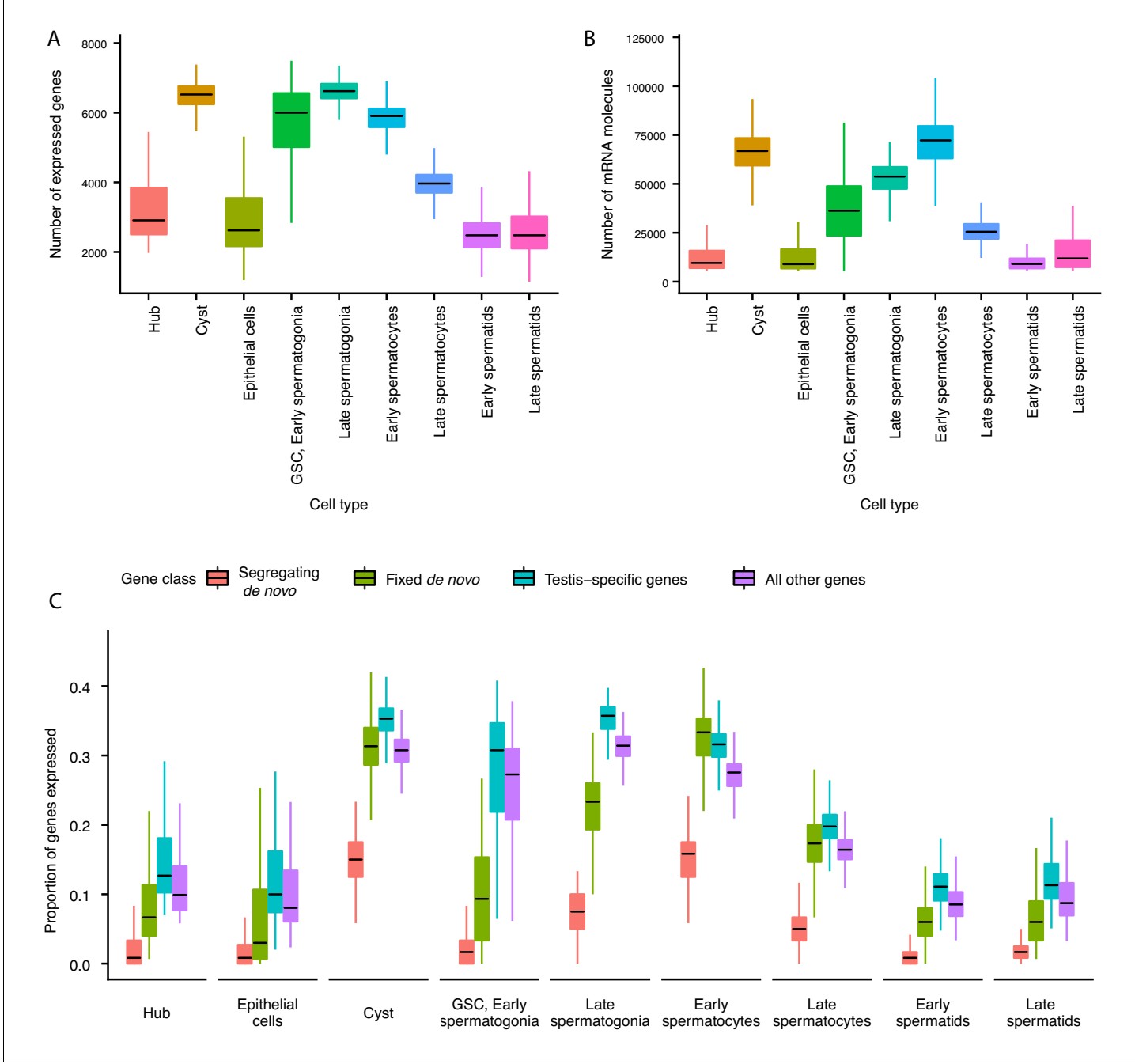

**Figure 2.** Gene expression and RNA content through spermatogenesis. (**A**) Boxplots of the number of genes expressed in each cell, binned by assigned cell type. Late spermatogonia and early spermatocytes express the most genes, and spermatids the least. (**B**) The number of Unique Molecular Indices (UMIs) detected for each cell, a proxy of RNA content. By this metric RNA content peaks in early spermatocytes, and is reduced thereafter by post-meiotic transcriptional suppression. (**C**) The proportion of segregating de novo, fixed de novo, testis-specific, and all genes expressed in every cell. For each cell, we counted the number of each class of gene with non-zero expression and divided it by the total number of genes of that type, grouping by cell type. For every cell type except spermatocytes, segregating de novo genes are the least commonly expressed, fixed de novo genes are more commonly expressed and all genes are most commonly expressed. In every cell type except early spermatocytes, a smaller proportion of fixed de novo genes are expressed than testis-specific genes, but early and late spermatocytes express similar proportions of fixed de novo genes and testis-specific genes. It is important to note that this measure looks at the number of genes of each type detected in a cell, not the expression level of each, and does not distinguish between high and low expression.

DOI: https://doi.org/10.7554/eLife.47138.006

After calculating the cell-type-specific expression profile for every detectable gene, we asked whether a given cell expresses similar proportions of de novo genes, testis-specific genes, and all other annotated genes. We observed that in most cell types, segregating de novo genes were the least commonly expressed group of genes, fixed de novo genes were more common, and testis-specific genes were most commonly expressed (*Figure 2C*). Early and late spermatocytes, however, express similar proportions of fixed de novo genes and testis-specific genes. Moreover, spermatocytes also show the highest relative abundance of segregating de novo genes compared to other cell types. Altogether, the high proportion of de novo genes expressed in spermatocytes suggests that such genes may play functional roles in these cells and development stage.

## Developmental trajectories show de novo gene expression bias during spermatogenesis

To study the transcriptomic path that a progenitor cell would take during its differentiation process, we reconstructed the developmental trajectory of spermatogenesis using monocle (*Trapnell et al., 2014*), which uses a graph-based minimum-spanning tree to align cells along an inferred path called pseudotime (*Figure 3A*, *Figure 3—figure supplement 1*). Pseudotime does not correspond to the actual timing of developmental processes; rather, it is a roadmap of cell differentiation as a function of transcriptomic changes. As an initial step to verify the accuracy of our pseudotime map, we plotted the number of UMIs detected as a function of pseudotime as a proxy of RNA content throughout spermatogenesis (*Figure 3B*). We saw that the number of UMIs starts fairly low, increases dramatically, and then decreases towards the end of pseudotime. This is consistent with the known post-meiotic downregulation of most transcription during spermatogenesis (*Barreau et al., 2008*).

By plotting inferred gene expression in every cell as a function of pseudotime, we approximated the behavior of individual genes throughout spermatogenesis. Marker genes consistently show a similar profile in pseudotime and the Seurat analysis. For instance, *bam*, *vas*, and *His2Av* enrichment denote the beginnings of spermatogenesis, and *fzo* and *twe* denote early and late spermatocytes, respectively (*Figure 3C*). Confident that our calculated pseudotime is an accurate representation of spermatogenesis, we proceeded to use it to query how the expression of de novo genes changes throughout spermatogenesis.

If a given novel gene is functional, one would expect it to be biased towards meiotic cells, since germline stem cell-specific genes may not undergo long-term and recurrent positive selection (*Choi and Aquadro, 2015*). If these genes confer limited beneficial effects, we predict that they may show stochastic transcription pattern in a large variety of cell types. Consistent with our predicted expression patterns of functional novel genes, we found that a large number of de novo genes are expressed specifically in a stage-biased manner, with a significant bias towards meiotic germ cells. Fixed annotated de novo genes show a variety of expression patterns over pseudotime (*Figure 3D*), with some showing bias towards early stages (*CG44174*), some with a bimodal expression pattern (*CG44329*), and some biased towards late spermatogenesis (*CR44412*). The top five most differentially expressed segregating de novo genes show a variety of expression patterns, but four of the five are biased towards early/middle pseudotime (*Figure 3E*).

## Gene age and mode of origination affects gene expression bias across cell types

Our prior observation that many de novo genes are enriched in GSC/early spermatogonia led us to ask whether the expression patterns of de novo genes differ from the expression patterns of other genes. Although individual de novo genes show a variety of expression patterns, we found that, compared to testis-specific genes, segregating de novo genes are less expressed in germline stem cells (p.adj = 9.35E-04) and slightly enriched in early spermatids (p.adj = 1.70E-02) (*Figure 4A*, *Table 1*). By contrast, the scaled expression of fixed de novo genes is not statistically different from that of testis-specific genes (*Figure 4B*, *Table 1*). These results suggest that cell-type expression patterns may impact the likelihood that a de novo gene will reach fixation.

We also asked whether this spermatocyte-biased expression is driven by segregating or fixed de novo genes. We quantified gene expression bias for segregating and fixed de novo genes separately and found that both groups of genes display the same direction of bias and a similar degree of statistical significance in every cell type (*Table 1*, *Supplementary file 3*). These results suggest that

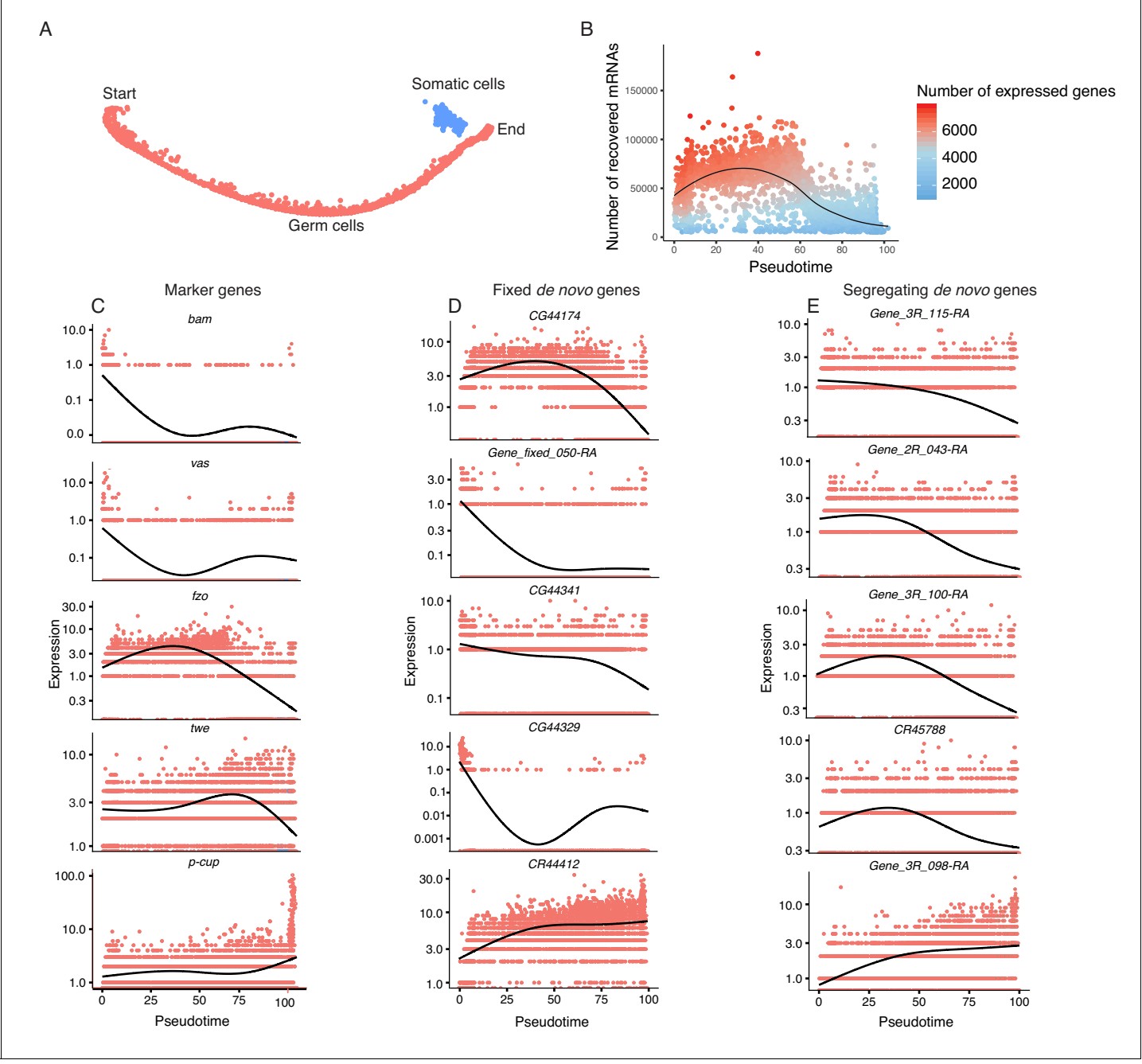

**Figure 3.** Pseudotime approximates the developmental trajectory of spermatogenesis. (**A**) We aligned every cell from our testis sample along an unsupervised developmental trajectory. From the expression of marker genes, we found somatic cells (blue) which were forced onto the developmental trajectory. For further analysis we disregard this branch (See Materials and methods, *Figure 3—figure supplement 1*). Spermatogenesis begins at the far-left end of the trajectory. (**B**) The relative RNA content per cell peaks in mid-spermatogenesis, and declines during spermatid maturation, as approximated by the number of UMIs detected per cell. The number of genes expressed declines as well. The black line is a Loess-smoothed regression of the data, which should be thought of as a general trend among stochastic data and not a mathematical model. (**C**) Loess-smoothed expression of marker genes along the red germ cell lineage assigned in panel A. Along this lineage, the relative expression of marker genes is consistent with their temporal dynamics inferred from previous work. (**D**) Fixed de novo genes show a variety of expression patterns, including biphasic, early-biased, and late-biased. (**E**) Segregating de novo genes are often biased towards early/mid spermatogenesis.

DOI: https://doi.org/10.7554/eLife.47138.007

The following figure supplement is available for figure 3:

**Figure supplement 1.** Assignment of somatic branch of pseudotime trajectory.

*Figure 3 continued on next page*

*Figure 3 continued*

DOI: https://doi.org/10.7554/eLife.47138.008

cell-type expression patterns do not impact the likelihood that a de novo gene will reach fixation, rather, the function and fitness effect may play an important role in the process of fixation.

Given a general trend for meiotic enrichment of de novo genes, we asked what proportion of de novo genes exhibit this pattern. Across pseudotime, we qualitatively estimated the relative expression biases of all de novo genes we could detect from *Zhao et al. (2014)*. Overall, 55% of segregating and 62% of fixed genes were biased towards middle stages (spermatocytes), and 11% of both segregating and fixed genes showed high expression bias toward later stages (spermatids). Surprisingly, 29% of segregating genes and 26% of fixed genes showed high expression bias toward early stages (GSC and spermatogonia) (*Figure 4C*). While many segregating de novo genes are highly expressed in early spermatogenesis, our results from *Figure 4A* suggest that as a group they are less expressed than typical testis-specific genes in GSC and early spermatogonia. This variety of expression patterns in young de novo genes indicates functional diversification in short evolutionary timescales.

Given that de novo genes, like typical testis-specific genes, are usually maximally expressed during meiosis, we asked if the expression dynamics of recently duplicated genes, another class of young genes, are similar (*Figure 4D*). Using a list of *D. melanogaster*-specific 'child' genes and their parental copies (*Zhou et al., 2008*), we queried expression of the parental and derived copies of duplicated genes over pseudotime (*Figure 4—figure supplement 1*). We classified gene expression patterns into 'early', 'late' 'middle' and 'bimodal' for each group. Only 2/14 'child' genes whose expression could be detected in testis had the same expression pattern as their parental copy, indicating that most derived gene copies are regulated by different mechanisms than their parental copy. All parental genes exhibited an early or late expression pattern, but child genes were a mixture of early, late, middle and bimodal expression patterns. (*Table 2*, *Figure 4—figure supplement 1*).

Bimodal expression (a peak in early and middle/late stages) is the most frequent expression pattern for child genes (43%), a pattern we did not observe for any parental genes. It is possible that these bimodal genes were originally expressed with the same pattern as their parental copy and later acquired expression in a different stage, consistent with neofunctionalization (*Ding et al., 2010*; *Lynch and Conery, 2000*).

We also observed strikingly different expression patterns for young genes depending on their mode of origination (duplication vs. de novo). To compare young genes of a similar age group, we quantified expression patterns for fixed *melanogaster*-specific de novo genes from *Zhao et al. (2014)*, and *melanogaster*-specific gene duplicates from *Zhou et al. (2008)*. We found that fixed de novo genes are most frequency biased towards mid-spermatogenesis (*Table 2*), and *melanogaster*-specific duplicate genes are most commonly bimodally expressed. This result indicates that a gene's expression pattern is influenced by its mode of origination. De novo genes often build regulatory sequences from scratch, but young gene duplicates may co-opt flanking promoter and enhancer sequences from their parental copy.

## Mutational load decreases throughout spermatogenesis

Since evolutionary innovations largely depend on novelties occurring at the DNA sequence level, we asked if the mutational load of germ cells varies during the process of sperm development. From our single-cell RNA-seq data, we identified 73 high-confidence substitutions that likely arose de novo. While the reference allele for every variant was present in somatic cells, the variant form of each of them was exclusively found in germ cells, and each inferred substitution is unlikely to be an RNA editing event or unrepaired transcriptional error (see Materials and methods). These substitutions were not present in population-level genome sequencing or previous whole-tissue RNA-seq of RAL517 testis, and the variant form of each substitution was also not present in any of our 3 types of somatic cells. We observed several instances of tightly clustered substitutions (<20 bp apart) present in the same cells, which we interpreted as single mutational events (*Supplementary file 2*, *Figure 5—figure supplement 1*). These substitution clusters could be the result of replicative errors

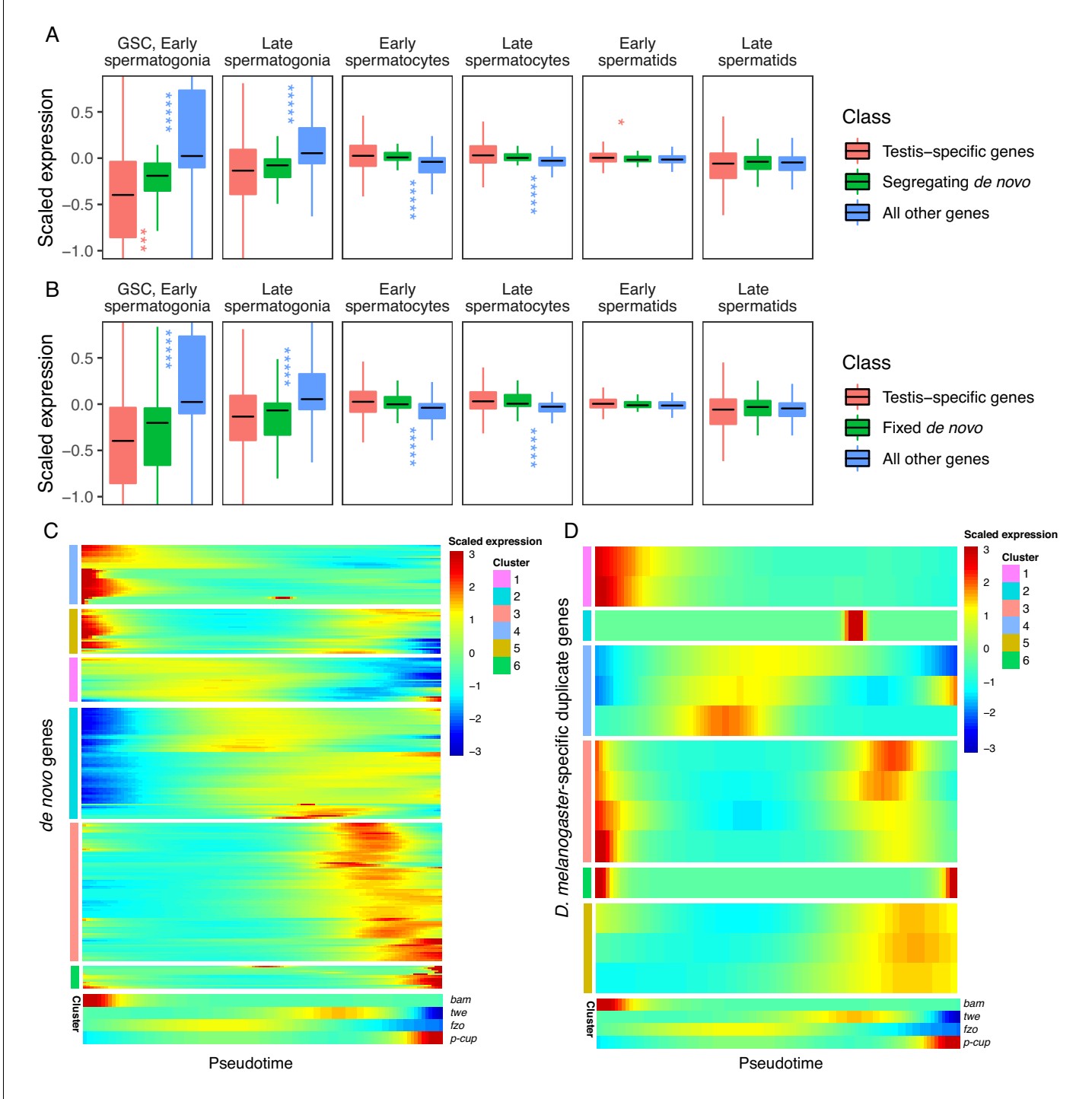

**Figure 4.** Expression bias of young genes. Spermatogenesis starts at GSC, early spermatogonia and proceeds rightward. (A) The scaled expression distribution of segregating de novo genes in each cell type, compared with the distribution of every other gene and testis-specific genes. For every gene, 0 is its mean scaled expression in a cell type, and the Y axis corresponds to Z scores of deviations higher or lower than that mean value. Asterixis represent Hochberg-corrected p values. The color of the asterixis indicates which gene set is being compared to de novo genes, and their placement above or below the boxplots indicates that gene set's relationship (higher or lower) to de novo genes. By this measure, de novo genes are biased downwards in early spermatogenesis and upwards in early spermatids. (B) The scaled expression patterns of fixed de novo genes are typical of testis-specific genes. (C) The scaled expression of detected fixed de novo genes across pseudotime (left to right), clustered by monocle's plot_pseudotime_heatmap function. While most de novo genes are biased towards intermediate cell-types, a small portion of de novo genes are most

*Figure 4 continued on next page*

*Figure 4 continued*
expressed during early and late spermatogenesis. (D) The scaled expression of *melanogaster*-specific duplicate genes over pseudotime. Despite being a similar evolutionary age to fixed de novo genes, young duplicate genes are more likely to be biased towards early and late spermatogenesis.
DOI: https://doi.org/10.7554/eLife.47138.009
The following figure supplement is available for figure 4:

**Figure supplement 1.** Expression heatmaps of parental and derived duplicated genes.
DOI: https://doi.org/10.7554/eLife.47138.010

resulting from the misincorporation of bi-nucleotides or multi-nucleotides, or due to the recruitment of an error-prone repair pathway at a double-strand break or bulky lesion. After counting clustered mutations as one mutational event, we obtained 44 mutational events present in one or more cell types (*Figure 5A*).

Putative de novo mutations are each likely unique to an individual. If a mutation were found in multiple individuals, it would likely be an inherited somatic variant and we would catch such mutated alleles in somatic cells. For each of the mutations, we identified reads from somatic cells with the WT allele at that position, and the mutated allele is only present in germ cells. Each variant is also supported by multiple germ cell reads with different UMIs.

To approximate per-base mutation load of each cell type, we accounted for two factors. Firstly, we are more likely to call a mutation event in more abundant cell types. Secondly, we are only able to detect mutation events in transcribed regions, so cells with a larger breadth of transcribed regions will likely yield more events. To control for these variables, we calculated the approximate per-base mutation load of each cell type by dividing the number of detected substitutions by the number of cells and the number of bases covered by 10 or more reads in that cell type, finding a decrease in the relative abundance of substitutions during the progression of spermatogenesis (*Figure 5B*).

Importantly, while we detected 30% (22/73) of inferred germline substitutions in early spermatids, we detected no germline variants in late spermatids. This means that either 1) most lesions are corrected by this stage, or 2) cells with lesions were removed by programmed cell death, or 3) that we captured insufficient quantities of mature spermatid mRNA to detect remaining variants. Although we found that early and late spermatids have similar RNA content, the low abundance of late

**Table 1.** Adjusted p values and direction of bias for gene expression biases of selected gene groups in germ cells.
Spermatogenesis progresses downward from GSC/Early spermatogonia and ends in late spermatids. Upwards arrows indicate that the top group of genes is biased upwards compared to the bottom group, and downwards arrows indicate that it is biased downward according to a directional Hochberg test. For example, ribosomal proteins are more expressed in late spermatogonia than all other genes, with an adjusted p value of 1.24E-75. Note that while segregating de novo genes are expressed differently from testis-specific genes in GSC, early spermatogonia and early spermatids, fixed de novo genes do not significantly deviate from expression patterns of testis-specific genes in any cell type.

| Versus: | Ribosomal protein genes | | Segregating de novo genes | | Fixed de novo genes | | DNA repair genes | |
|---|---|---|---|---|---|---|---|---|
| | All other genes | Testis-specific genes | All other genes | Testis-specific genes | All other genes | Testis-specific genes | All other genes | Testis-specific genes |
| GSC, early spermatogonia | ↑ 1.13E-82 | ↑ 1.44E-84 | ↓ 1.46E-21 | ↑ 9.35E-04 | ↓ 2.92E-22 | ns | ↑ 4.69E-26 | ↑ 8.14E-62 |
| Late spermatogonia | ↑ 1.24E-75 | ↑ 1.62E-74 | ↓ 8.22E-19 | ns | ↓ 5.89E-18 | ns | ↑ 2.30E-20 | ↑ 5.80E-44 |
| Early spermatocytes | ↓ 2.53E-76 | ↓ 1.08E-71 | ↑ 4.08E-15 | ns | ↑ 5.31E-13 | ns | ↓ 6.50E-23 | ↓ 1.75E-38 |
| Late spermatocytes | ↓ 2.51E-57 | ↓ 1.90E-58 | ↑ 1.09E-10 | ns | ↑ 2.17E-15 | ns | ↓ 3.96E-09 | ↓ 4.58E-29 |
| Early spermatids | ↓ 8.94E-03 | ↓ 1.57E-08 | ns | ↓ 1.70E-02 | ns | ns | ↑ 5.89E-08 | ns |
| Late spermatids | ↓ 7.40E-10 | ↓ 1.70E-02 | ns | ns | ns | ns | ns | ns |

DOI: https://doi.org/10.7554/eLife.47138.011

**Table 2.** Frequency of pseudotime expression patterns for *melanogaster*-specific fixed de novo genes and *melanogaster*-specific duplicate genes.

For genes in **Figure 4C and D**, we counted the number of genes showing a strong bias for early pseudotime, late pseudotime, mid-pseudotime, or a bimodal expression pattern. Fixed de novo genes are most frequently biased towards mid-pseudotime and the plurality of *melanogaster*-specific child duplicate genes show a bimodal expression pattern. Pseudotime expression plots of the parent-child duplicate gene pairs used in this analysis are in **Figure 4—figure supplement 1**. Proportions are rounded to two decimal places and may not add up to 1.

| Pattern | Fixed de novo proportion | Parental duplicate proportion | *melanogaster*-specific child duplicate proportion |
|---|---|---|---|
| Early | 0.26 | 0.37 | 0.14 |
| Mid | 0.62 | 0.00 | 0.21 |
| Late | 0.11 | 0.63 | 0.21 |
| Bimodal | 0.01 | 0.00 | 0.43 |

DOI: https://doi.org/10.7554/eLife.47138.012

spermatids makes either explanation possible. Since we observed a steady downward trend of mutation abundance during the progression of spermatogenesis, it is reasonable to infer that late spermatids have a mutational burden equal to or less than that of early spermatids. We counted the number of cells of each type carrying mutations throughout spermatogenesis. We observed that the relative proportion of cells carrying mutations drops consistently throughout spermatogenesis (**Figure 5C**), indicating that mutational load decreases during spermatogenesis. A chi-square test of the trend in proportions shows that the relative numbers of mutated cells follow a linear trend (p value = 2.20E-16). This result is highly statistically significant and lends credence to our other observations of dwindling mutational load during spermatogenic progression. This trend could be the result of active lesion repair, or the death of cells carrying unrepaired lesions.

## DNA repair genes and ribosomal protein genes show an early expression bias

We asked whether two key programs, DNA repair, and translation, show signatures of expression bias during spermatogenesis. We hypothesized that both programs are critical to the production of healthy spermatids, which must undergo heavy periods of growth and division without accumulating mutations.

Ribosomal protein genes appear to be strongly biased towards early spermatogenesis (**Table 1**, **Supplementary file 3**). Compared to testis-specific genes or all other genes, they are upregulated in GSCs/early spermatogonia and late spermatogonia, and downregulated in early spermatocytes, late spermatocytes, early spermatids, and late spermatids. Our results indicate that translation is required at the very beginning of spermatogenesis, possibly to build cellular machinery during a period of rapid growth and division. Interestingly, recent studies suggest that ribosomes play an important role in regulating stem cell fate and homeostasis (**Nagy et al., 1993**; **Turner, 2008**). The observed abundance of those ribosome protein genes is consistent with ribosome loading playing an important role in stem cell differentiation and germ cell differentiation. Translation is important for later spermatogenesis as well, and our results indicate that the ribosomal machinery may be built early and stored for use in later developmental stages.

We hypothesized that since replication and transcription are very active in early spermatogenesis, DNA repair-gene expression may be biased towards early spermatogenesis. We quantified the expression pattern of 211 DNA repair genes in the testis (DNA repair genes were taken from **Svetec et al., 2016**). We found that, compared to testis-specific genes, DNA repair genes were upregulated in GSCs/early spermatogonia and late spermatogonia (corr. p values = 8.14E-62, 5.80E-44, respectively), and depleted in early and late spermatocytes (corr. p values = 1.75E-38, 4.58E-29, respectively) (**Figure 5D**, **Table 1**). We reason that DNA repair genes are transcribed early because the DNA repair machinery must be assembled early in order to repair mutations as soon as they occur.

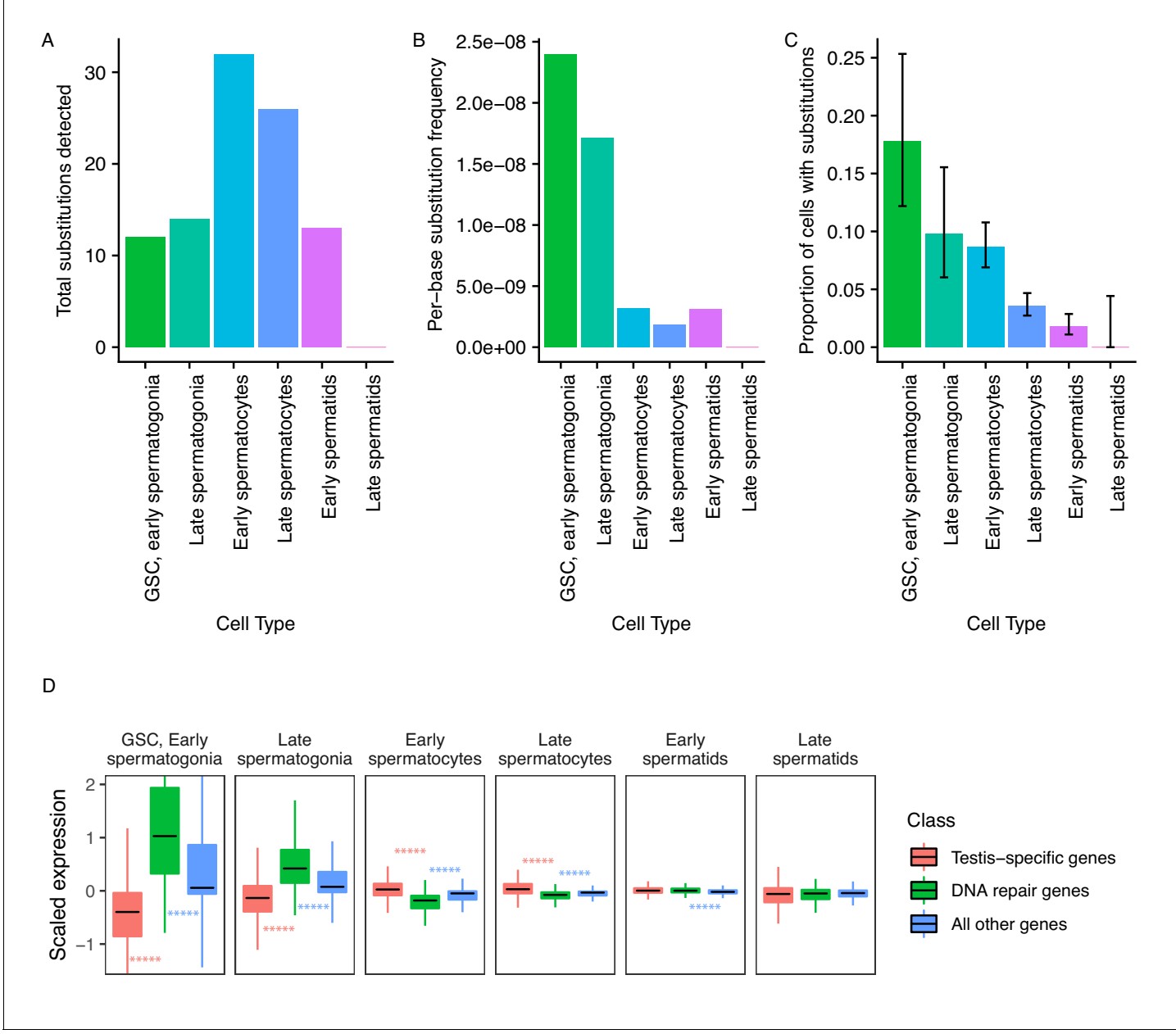

**Figure 5.** Abundance of putative de novo germline mutations. (**A**) For every cell type, the total number of high-quality polymorphisms identified. Out of 2590 candidate variants, we excluded all substitutions that could be found in any somatic cell, leaving 73 variants. We then counted clustered polymorphisms as single mutational events and removed variants that could have resulted from RNA editing. See Materials and methods for details. (**B**) Dividing the number of polymorphisms in a cell type by the number of cells of that type, and the number of bases covered with at least 10 reads in that cell type (*Supplementary file 5*) yields an approximate relative substitution frequency for each cell type. By this metric, substitutions are most prevalent in early spermatogenesis, and decrease in relative abundance during spermatid development. This could be due to the apoptosis of mutated cells, or the systematic repair of DNA lesions during spermatogenesis. (**C**) The proportion of cells of each type with at least one identified germline lesion. Error bars are the 95 percent confidence intervals for each proportion. A Chi-square test for trend in proportions gives a p value of 2.20E-16, indicating strong evidence of a linear downward trend. (**D**) DNA repair genes are generally biased towards early spermatogenesis, statistically enriched compared to the distribution of all other genes. (Wilcoxon adjusted p value < 0.05).
DOI: https://doi.org/10.7554/eLife.47138.013

The following figure supplement is available for figure 5:

**Figure supplement 1.** Alignment of germline mutations along pseudotime.
DOI: https://doi.org/10.7554/eLife.47138.014

## Discussion

Our findings provide an unprecedented perspective on evolutionary novelty within the testis. We have developed a simple but robust method to quantify gene expression bias in a cell-type specific manner in single-cell data. It revealed the presence of expression biases in DNA repair genes, segregating de novo genes, and other gene groups. Zhao et al. (2014) characterized de novo genes as lowly expressed from bulk RNA-seq data, but our data demonstrate that de novo genes show various expression patterns among all cell types and are commonly expressed in spermatocytes. Our other observation that segregating de novo genes exceed the expected post-meiotic expression of testis-specific genes is also intriguing. One possibility is that some de novo transcripts may escape RNA degeneration and have a long lifespan after meiosis. Over time, if the products of de novo transcripts are selected and modified by natural selection, the regulatory sequences and resulting expression pattern will be refined. Since fixed de novo genes show similar scaled expression patterns to older testis-specific genes, it is possible that certain expression patterns common to older testis-specific genes increase the likelihood of a segregating gene reaching fixation, or that many of the de novo genes function similarly to testis-specific genes.

Our finding that young duplicated genes have different expression patterns than de novo genes merits further study. Young duplicated genes are more likely to be bimodally expressed than de novo genes of a similar age. While it appears that de novo genes are often highly expressed during meiosis, many duplicated genes display a bimodal expression pattern. While de novo genes may have relatively simple minimal promoters, young duplicated genes may maintain much of the regulatory sequence of their parent copy. However, since only 2 out of 14 melanogaster-specific child duplicates have the same general expression pattern as their parent genes, it seems that their regulatory sequences are often modified to produce alternative expression patterns. This observation suggests that some young duplicated genes have relaxed selective pressure to perform the ancestral function, allowing for neofunctionalization.

We also developed a method to extract mutational information from single-cell RNA-seq data, which can provide information about germline or de novo DNA lesions present in a sample without the need for DNA sequencing. While our method cannot identify variants in untranscribed regions, introns, or sense strands, our method approximates the relative mutational load of different cell types in a sample. Variation in RNA content between cell types may decrease our power to detect substitutions in less transcriptionally-active cells such as late spermatids, although our calculated mutational load in *Figure 5B* accounts for this. Despite the lack of data for late spermatids, our results suggest that many errors are at least partially repaired before the completion of spermatid maturation. Alternatively, cell death could have removed mutated cells before spermatid maturation if a lesion could not be repaired. Our data cannot distinguish between the death of mutated cells and successful repair of lesions.

The variants we have found are either DNA lesions that have escaped repair, or the lesions that have been selected through competition among cells (Loewe and Hill, 2010). It is likely that some of these substitutions would result in inviable offspring and would never be observed in an adult population. Our result suggests that mutational load varies between different cell divisions, consistent with previous work that suggests a variable lesion rate between cell types (Gao et al., 2011; Gao et al., 2014). Mutational load is the net product of damage and repair, and further characterization of how lesions occur and accumulate in the germline is needed to better understand the evolutionary ramifications of this process (Moorjani et al., 2016).

It appears that the cell types with the highest mutational load are germline stem cells/early spermatogonia, the earliest germ cells. This indicates that by the time germline stem cells enter spermatogenesis, they carry a relatively high mutational burden. This could be due to the fact that germline stem cells cycle many times, dividing asymmetrically to produce spermatogonia and a replenished germline stem cell. This cycle, repeated enough times, could cause a buildup of variants in germline stem cells as they continue to produce spermatogonia. Such a scenario would necessitate a mechanism to remove high level of lesions from maturing gametes. This mechanism must be an equilibrium removing enough lesions to prevent the accumulation of harmful phenotypes. However, the population-level mutation rate never reaches zero, otherwise adaptive evolution will cease (Lynch, 2010).

Since we observed that the transcription of 211 DNA repair genes drops during meiotic stages, we suspect that DNA repair gene products are translated early and continue to repair lesions throughout spermatogenesis. After meiosis, however, the gametes become haploid, and there is no longer a template strand to facilitate homology-directed repair. This should constrain the types of DNA damage repair available in late spermatogenesis. It is also important to note that male *Drosophila* do not undergo meiotic recombination, meaning that the DNA repair events that occur during spermatogenesis are likely due to replicative or transcriptional forces, not recombination. Transcription-coupled repair during spermatogenesis is apparent in mouse and humans, as variants on the template strand and the coding strand of testis-expressed genes are asymmetrical (*Xia et al., 2018*). Our finding that the number of variants decreases throughout spermatogenesis is consistent with the results of Xia et al., who posit a generalized genomic surveillance function of spermatogenesis. Future work should use single-cell genome sequencing on FACS-purified subpopulations of testis cells to identify germline variants and calculate their relative abundance. Additionally, our work necessitates comparison of the relative mutational burden of older flies to younger flies. If DNA lesions accumulate in cycling germline stem cells over time, spermatogenic mutational surveillance may less efficiently compensate for more lesions in sperm from older individuals. Our result indicates that cell-type specific mutational load can be estimated from single-cell RNA-seq data with reasonable accuracy. Overall, we provide novel insights into the dynamics of mutation, repair, and de novo gene expression profiles in the male germline.

## Materials and methods

### Key resources table

| Reagent type (species) or resource | Designation | Source or reference | Identifiers | Additional information |
|---|---|---|---|---|
| Strain, strain background (*Drosophila melanogaster*, male) | RAL517 | *Mackay et al., 2012* | BDSC:25197 | |
| Commercial assay or kit | 10X chromium 3' kit V2 | 10X genomics | 10X genomics product number CG00052 | |
| Chemical compound, drug | Gibco Collagenase, type I | Thermo Fisher | Thermo Fisher catalog number 17018029 | |
| Chemical compound, drug | Trypsin LE | Thermo Fisher | Thermo Fisher catalog number 12605036 | |
| Software, algorithm | Cellranger | 10X genomics | | |
| Software, algorithm | Hisat2 | PMID:25751142 | | |
| Software, algorithm | Stringtie | PMID:25690850 | | |
| Software, algorithm | Seurat | *Satija et al., 2015* | | |
| Software, algorithm | bcftools | PMID:28205675 | | |
| Software, algorithm | samJDK | *Lindenbaum and Redon, 2018* | | |
| Software, algorithm | Monocle | *Trapnell et al., 2014* | | |

### Preparation and sequencing of testis single-cell RNA-seq libraries

We used 2- to 3- day-old DGRP-RAL517 flies in this study (*Mackay et al., 2012*). Testes from 50 male flies were dissected in cold PBS. The resulting 50 testes were de-sheathed in 200 µl of lysis buffer (Trypsin LE + 2 mg/ml collagenase). The samples were incubated in lysis buffer for 30 min at

room temperature with gentle vortex mixing every 10 min. The samples were filtered through a 30 µm tissue culture filter followed by a 7 min centrifugation at 1200 rpm. The cells were washed with 200 µl of cold HBSS and pelleted again for 7 min at 1200 rpm. The resulting cell preparation was re-suspended in 20 µl of HBSS before further processing. For cell counting, 5 µl of the single cell suspension were mixed with 5 µl of the exclusion dye trypan blue and the total cell number as well as the ratio between live and dead cells were analyzed using an automated cell counter (Logos Biosystems). For imaging, 15 µl of the cell suspension were transferred to a slide and imaged in a Zeiss upright light microscope. This method yielded high numbers of single cells with an average of 93–96% viability. We then submitted 8000 cells (sequenced 5000 cells) for library preparation with the 10X Genomics chromium 3' kit, followed by sequencing with Illumina Nextseq 98 bp paired-end chemistry.

## Preparation of custom annotation file for de novo gene analysis

We analyzed de novo genes identified in *Zhao et al. (2014)*, by converting the gene coordinates to *D. melanogaster* version six reference genome with FlyBase coordinates converter. Strand data and splicing information is not present for those reference genes, so we chose to proceed only with genes whose expression could be detected in our *D. melanogaster* testis single-cell sequencing data. Using whole-tissue RNA-seq data from multiple strains of *Drosophila* testis, we used Stringtie merged to create a merged transcriptome GTF containing unannotated transcripts and used BLAST to compare the novel transcripts against converted coordinates for the *Zhao et al. (2014)* genes. For genes with a match between the converted 2014 coordinates and the new merged transcriptome, we added the coordinates from the merged GTF to the FlyBase dmel_r6.15 reference GTF. Since a single-exon de novo genes could be on either strand, we created a plus and minus strand version of every verified de novo gene. Our custom annotation file thus contains all the standard FlyBase dmel_r6.15 genes, plus a set of assembled transcripts known to correspond to de novo genes.

Our study only seeks to analyze previously characterized de novo genes, and will inherit the limitations of identification of de novo genes using bulk RNA-seq data. *Zhao et al. (2014)*, the source paper for these segregating and fixed de novo genes, detected de novo genes from bulk testis RNA-seq of multiple *D. melanogaster* strains, meaning that de novo genes that are enriched in a rare cell type may not be counted as de novo genes if their expression in the whole tissue does not reach a certain threshold. Despite this possibility we still observe many de novo genes with maximum expression in rare cell types such as germ line stem cells and spermatogonia.

## Quantification of reproducibility

If single-cell suspension results to relatively unbiased ratios of cell types compared to in vivo cell types, one would expect a relatively high correlation of single-cell RNA-seq and bulk RNA-seq data. To verify this, we aligned the single-cell RNA-seq reads and bulk RNA-seq reads of RAL517 separately to the reference genome using Hisat2, calculated gene TPMs with Stringtie, and then used DEseq2 to regularized-log transform the TPM values from both datasets. After that, we plot the correlation of normalized gene expression and calculated the Pearson's R (*Figure 1—figure supplement 2A*). Using the same method, we also compared our dataset to a second single-cell library prepared from testis of a wild *D. melanogaster* strain from our lab (*Figure 1—figure supplement 2B*).

## Processing of single-cell data

Illumina BCL files were converted into fastq files using Cellranger mkfastq. A reference genome was created with Cellranger mkref, with all genes from the FlyBase *D. melanogaster* reference. To this reference, we added all segregating and fixed de novo genes from *Zhao et al. (2014)*. We used the custom reference to run Cellranger count, which demultiplexed the single cell reads into a usable format for Seurat. Going forward, we kept all genes expressed in at least 3 cells and all cells with at least 200 genes expressed. We ran Seurat ScaleData and NormalizeData with default parameters. According to the Seurat documentation, 'Feature counts for each cell are divided by the total counts for that cell and multiplied by the scale.factor (default = 10,000). This is then natural-log transformed using log1p.' We then ran Seurat's default t-SNE function and found clusters based on the first nine principal components (resolution = 2). Of the parameters we tried, most produced a similar t-SNE

clustering pattern, but nine principal components generated the best separation between different cell types.

## Identification of cell types in single-cell RNA-seq data

We used marker genes to infer the predominant cell type within each cluster in Seurat. *Aubergine* (*aub*) is a marker of germline stem cells (*Rojas-Ríos et al., 2017*), and Bag of Marbles (*bam*) is a marker of spermatogonia (*Kawase et al., 2004*). A cluster enriched in *vasa* (*Ohlstein and McKearin, 1997*) and *bam* but not *aub* was annotated as late spermatogonia. Clusters most enriched in *fuzzy onions* (*fzo*) were inferred to be early spermatocytes (*Hwa et al., 2002*), and clusters with enrichment of twine (*twe*) but not *fzo* were inferred to be late spermatocytes (*Courtot et al., 1992*). The literature is clear that transcription of *fzo* and *twe* peaks in spermatocytes, but it is less clear which marker denotes early and late spermatocytes, respectively. To resolve this ambiguity, we used monocle (*Trapnell et al., 2014*) to align our cells on a developmental trajectory called pseudotime (rho = 68, delta = 5, ordered using the top 1000 differentially expressed genes). We found that *twe* expression peaked later in spermatogenesis than *fzo*, and concluded that clusters expressing *twe* but not *fzo* were late spermatocytes. Epithelial cells were defined based on enrichment of *MntA* and *Hsp23*, Hub cells were defined based on *Fas3*, and Cyst cells were defined by enrichment of *zfh1* (*Zhao et al., 2010*). Late spermatids were marked by *p-cup*, a post-meiotically transcribed gene.

## Analysis of the spermatogenic developmental trajectory

The adult testis contains both somatic and germ cells, but lacks the common progenitor cells for each lineage. Therefore, when constructing a lineage tree for all cells in our tissue, we would expect to see a separate branch containing somatic cells erroneously branching from somewhere along the inferred lineage of more common germ cells. In the somatic cell branch from *Figure 3A*, *MtnA* is enriched (*Figure 3—figure supplement 1*), leading us to infer that this state is mainly somatic cells. As such, we ignored this branch for our analysis of gene expression during germ cell development in *Figure 3*. One should not interpret this result as evidence that somatic cells in the testis arose from germ cell progenitors, rather, this is a consequence of Monocle's algorithm that forces a minimum spanning tree for all cells in a sample, regardless of their real cell-type of origin. Since the original common progenitor for the germ lineage and somatic testis cells is not present in adult tissues, Monocle placed the somatic cells to their closest germ cell neighbors, in this case late germ cells. As shown in *Figure 3*, there is a gap between the group of somatic cells and the tightly clustered lineage of germ cells, indicating that the cells are indeed from a different lineage.

To construct the trajectory, we used the following parameters:

- >reduceDimension(max_components = 2, num_dim = 3, norm_method = 'log', reduction_method = 'tSNE')
- >clusterCells(my_cds, rho_threshold = 55, delta_threshold = 10, frequency_thresh = 0.1)

To order the cells, we used the top 1000 genes with the highest q value of being differentially expressed between clusters.

## Calculation of cell-type bias of gene groups

Testing whether gene expression is biased across cell types requires overcoming two challenges. Firstly, different cell types have varying levels of RNA and global transcription, so it is important to account for the behavior of other genes in a cell type when calculating expression bias of a group of genes. Additionally, the calculated expression values for different groups of genes will vary by orders of magnitude. Expression values must be scaled across the dataset on a per-gene basis, with 0 representing a gene's mean expression across the tissue, and positive or negative values corresponding to the Z-score of a calculated expression value. To address the confounding effect of global variation in gene expression, we compared groups of genes against all other genes within a cell type, and asked if some groups of genes behave as outliers in a given cell type. For de novo genes, we compared the scaled, average expression of putative de novo genes to every other gene within a cell type using a signed Wilcoxon test (*Wilcoxon, 1945*).

For groups of genes (e.g. de novo genes, DNA repair genes), we asked whether their scaled expression distribution in a cell type was statistically different from that of other genes. For every gene, we calculated its average scaled expression within each cell type, and then performed a

Wilcoxon signed test to determine if the mean scaled expression of genes in the cell type was statistically higher or lower than all other genes in the same cell type. For each gene group and cell type, we adjusted the resulting p-values with Hochberg's correction (*Haynes, 2013*). This shows the direction and statistical significance of each cell-type specific bias of a gene group. For the raw and adjusted p values of every gene group tested, please refer to *Supplementary file 3*. For germline cells, the direction of bias and adjusted p values are given in *Table 1*. Gene lists used are in *Supplementary file 6*.

## Calculation of base substitution rate for individual cells

Using the demultiplexed, aligned reads generated by Cellranger, we ran bcftools mpileup (*Narasimhan et al., 2016*) with a minimum quality cutoff of 25 to find nucleotide polymorphisms from our RNA-seq data. We filtered the calls to exclude variants known to be segregating in populations of *D. melanogaster* DGRP-RAL517 (*Mackay et al., 2012*). We also filtered the variant calls against a *D. melanogaster* DGRP-RAL517 population genome dataset we generated recently. We also excluded variants whose read coverage for the reference allele was less than 10. With the remaining 2590 polymorphisms, we used samjdk (*Lindenbaum and Redon, 2018*) to extract reads containing the variant allele and match the cell barcode to the cell identities from our Seurat analysis. To remove variants that likely arose prior to the collection of this data, we excluded variants found in somatic cells (hub, cyst, epithelial cells). The numbers of variants remaining after each filtering step is given in *Supplementary file 4*.

We found a number of substitutions clustered together in close proximity and expressed in the same cells (*Figure 5—figure supplement 1*). We treated these clusters as single mutational events to avoid biasing our calculated mutational abundance. After counting the total variants detected within each cell type, we subtracted polymorphisms found within 10 bp of each other in the same cells so that each cluster of variants counted as one mutation event. To approximate cell-type specific substitution rate, the number of mutational events detected in each cell type was divided by the number of cells and the number of bases covered by at least 10 reads by all cells of a type using samtools. Number of cells, mutational events and covered bases are given in *Supplementary file 5*.

To ensure that our inferred mutations are not uncorrected transcriptional errors, we made sure each variant followed at least 2 of following criteria: (1) The alternate allele for most of our inferred mutations was found in multiple germ cells (but not somatic cells). A transcriptional error is unlikely to happen at the same position in multiple cells. (2) In every cell where a mutation was identified, the reference allele was either completely absent (possible homozygote) or present with as many or fewer reads as the alternate allele (possible heterozygote). (3) For substitutions found in only one cell, the alternate allele was present on multiple mRNA molecules (different UMIs). A transcriptional error is unlikely to produce the same change at the same position multiple times.

We performed the following steps to remove possible RNA editing events from our samples. Recurrent RNA editing events would be present in whole-tissue RNA-seq data, so we ran bcftools mpileup with the same parameters on whole-tissue testis RNA seq data of *D. melanogaster* RAL 517. Four of our seventy-seven inferred germline variants were present in the whole-tissue data, so we removed them for downstream steps. The final list does not show a high level of A > G substitution, which would be expected from RNA editing (*Tan et al., 2017*).

## Calculation of the proportion of mutated cells, by type

We manually checked every SNP with every cellular barcode within which the alternate allele was found. Using cellular identities that we inferred using Seurat, we counted the number of cells of each type containing at least one substitution. This number, divided by the total cells identified as that type, yields the proportion of mutated cells shown in *Figure 5C* and *Supplementary file 5*.

## Acknowledgements

We thank Connie Zhao and Nneka Nnatubeugo for the help on the single-cell sequencing experiment. We thank Kristofer Davie for the suggestions on single-cell suspension. We thank members of the Zhao laboratory for helpful discussions during the work. We are grateful to Mia Levine, Leslie Vosshall, David Begun, Ziyue Gao, Molly Przeworski, Xiaolan Zhao, and Sohail Tavazoie for critical reading of an earlier version of the manuscript.

## Additional information

### Funding

| Funder | Grant reference number | Author |
|---|---|---|
| Robertson Foundation | | Li Zhao |
| Monique Weill-Caulier Trust | Monique Weill-Caulier Career Scientist Award | Li Zhao |
| Alfred P. Sloan Foundation | Research Fellowship | Li Zhao |

The funders had no role in study design, data collection and interpretation, or the decision to submit the work for publication.

### Author contributions

Evan Witt, Data curation, Software, Formal analysis, Validation, Investigation, Visualization, Methodology, Writing—original draft; Sigi Benjamin, Resources, Investigation, Methodology; Nicolas Svetec, Conceptualization, Resources, Investigation, Methodology, Writing—original draft; Li Zhao, Conceptualization, Resources, Data curation, Software, Supervision, Funding acquisition, Writing—original draft

### Author ORCIDs

Evan Witt (iD) https://orcid.org/0000-0003-2973-6946
Sigi Benjamin (iD) https://orcid.org/0000-0002-6411-5339
Nicolas Svetec (iD) https://orcid.org/0000-0001-9617-2752
Li Zhao (iD) https://orcid.org/0000-0001-6776-1996

### Decision letter and Author response

Decision letter https://doi.org/10.7554/eLife.47138.027
Author response https://doi.org/10.7554/eLife.47138.028

## Additional files

### Supplementary files

• Supplementary file 1. Table of the 50 most enriched genes within cell types. Calculated by Seurat, positive markers only, ranked by average fold change.
DOI: https://doi.org/10.7554/eLife.47138.015

• Supplementary file 2. Top 10 enriched Gene Ontology (GO) terms in every cell type. For every gene that Seurat deemed enriched in a given cell type with a q value < 0.05, we queried the ID's against the PANTHER web database for *Drosophila melanogaster* (default parameters) and kept the top 10 enriched GO terms with the highest fold change and Bonferroni-adjusted p value < 0.05.
DOI: https://doi.org/10.7554/eLife.47138.016

• Supplementary file 3. Mathematical comparisons of gene bias between cell types for various gene groups. Corresponding to *Figures 4* and *5*, and *Table 1*, this table indicates the raw and Hochberg-adjusted p. values comparing each gene group's scaled expression distribution to the scaled expression distribution of testis-specific genes and all other genes within a cell type. P.greater is the p value for the gene set being expressed higher than the control set, and p.less is the p value for the gene set being expressed less than the control set in the cell type. Hochberg-corrected p values are the final two columns in each table. For example, in early spermatids, de novo genes have a p of 2.47E-04 and an adjusted p. value of 2.72E-03 to have higher scaled expression than testis-specific genes. A simplified version of this data is presented in *Table 1*.
DOI: https://doi.org/10.7554/eLife.47138.017

• Supplementary file 4. Filtering steps for Single Nucleotide Polymorphism calls. The 44 variants remaining at the end of the process were considered candidates for de novo germline mutations, since the reference allele is present in the population but the mutant allele is only present in germ-line cells.

DOI: https://doi.org/10.7554/eLife.47138.018

• Supplementary file 5. Counts of Single Nucleotide Polymorphisms per cell type. 'Polymorphisms detected' is the raw values for *Figure 5A*. Included for each cell type is the mean number of genes expressed and the number of cells of that type, allowing the calculation of variants/cell/covered base in *Figure 5B*. This table also contains, for each cell type, the number of cells with detected mutations. This is used to calculate the proportion of mutated cells in *Figure 5C*.
DOI: https://doi.org/10.7554/eLife.47138.019

• Supplementary file 6. Gene lists used to compare scaled expression bias of gene groups. For gene groups mentioned in *Figures 4* and *5*, these lists are the genes used.
DOI: https://doi.org/10.7554/eLife.47138.020

• Transparent reporting form
DOI: https://doi.org/10.7554/eLife.47138.021

### Data availability

Fastq files of the single-cell testis RNA-seq data have been deposited at NCBI SRA with accession numbers SAMN10840721 (RAL517 strain in main text, BioProject # PRJNA517685) and SAMN12046583 (Wild strain used for reproducibility analysis in Figure 1-figure supplement 2, PRJNA548742). Script used to create the custom reference and run the cellranger pipeline is available at https://github.com/LiZhaoLab/2019_Dmel_testis_singlecell (copy archived at https://github.com/elifesciences-publications/2019_Dmel_testis_singlecell), along with the custom reference used for the analysis.

The following datasets were generated:

| Author(s) | Year | Dataset title | Dataset URL | Database and Identifier |
|---|---|---|---|---|
| Evan W, Benjamin S, Svetec N, Zhao L | 2019 | D. melanogaster testis single-cell sequencing | https://www.ncbi.nlm.nih.gov/bioproject/?term=PRJNA517685 | NCBI BioProject, PRJNA517685 |
| Evan W, Benjamin S, Svetec N, Zhao L | 2019 | D. melanogaster testis single-cell sequencing | https://www.ncbi.nlm.nih.gov/bioproject/?term=PRJNA548742 | NCBI BioProject, PRJNA548742 |

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
