## [Decision Letter]

Thank you for submitting your article "Mutational and transcriptional dynamics across *Drosophila* spermatogenesis at the single-cell level" for consideration by *eLife*. Your article has been reviewed by three peer reviewers, one of whom is a member of our Board of Reviewing Editors, and the evaluation has been overseen by Patricia Wittkopp as the Senior Editor. The following individual involved in review of your submission has agreed to reveal her identity: Helen White-Cooper (Reviewer #3).

The reviewers have discussed the reviews with one another and the Reviewing Editor has drafted this decision to help you prepare a revised submission.

Summary:

This is a very interesting study that reports single cell mRNA sequencing on male *Drosophila* gonads to look at the spermatogenesis developmental program. The authors also examine the expression timing of young genes that evolved de novo or that duplicated recently. The reviewers were enthusiastic about the work but they raised major concerns that would need to be addressed. All of these concerns except one require only partial re-analysis et re-interpretation of the results. One of them concerns the integrity of the material subjected to single-cell sequencing. Although we understand that it is not possible to repeat the entire protocol, we agreed on the fact that a critical assessment of the staging and assignment to categories was required. We would also like to see a repeat of the cell preparation so that the reader can have more confidence in what they are seeing for different cell types.

Essential revisions:

1) For non-drosophilist readers, it is very important that the authors explain early on that male *Drosophila* do not typically undergo meiotic recombination. This has important implications for the interpretation of some of the results (especially the section on mutational load).

2) The single cell sequencing experiments were conducted on a pool of testes from 50 flies. We can appreciate that single cells are being profiled and this is some kind of within experiment replication, but this does not provide us with information on how reproducible the entire experiment is. The authors may have access to more data or independent data that would allow to demonstrate the reproducibility of the analyses. If it is the case, it would be useful to provide an analysis of such data. In addition, it is unclear to me how single-cell RNA-seq data is normalized (or not) and what the variance level is among cells. This seems like an important point that needs to be considered (whether the authors think this is a problem or not). If spermatocytes have overall higher transcriptional activity than other cell types (which could be the case considering more genes and mRNA molecular are detected, Figure 2A,B), this could in turn increase power to detect de novo genes, especially if these tend to be expressed at low levels (and affect some conclusions drawn, e.g. de novo genes to be expressed in higher proportion at the early spermatocyte stage, Figure 2C).

3) The identification of de novo mutations needs to be supported or discussed. Two reviewers were convinced but another one raised these points. It is surprising that many mutations are identified. Also, the analysis needs to be refined. The tissues examined are from a pool of flies. Are mutations from different individuals? 73 substitutions means about 1.5 substitution per fly since 50 flies are studied. Are mutations identified at one stage carried on to the next stage in the data? The per-base mutation rate would need to take into account the number of cell divisions, no? Are all stages going through cell division?

4) Could the authors make a prediction as to what active lesion repair vs death of cells carrying unrepaired lesions would look like in their data? I think the latter is more likely than the former given the observations (and that this is a very important and exciting result that could stand out more), because I predict that lesions would be difficult to transcribe (and perhaps lead to various nt misincorporation?), but I could be wrong. If correct, this result has important implications with respect to selective pressure within the testis.

5) The critical point of this paper in terms of determining whether the conclusions are supported by the data is the generation and interpretation of the single cell sequencing data, and one expert reviewer remains to be convinced by all of it. She would be much more reassured that the cells are healthy and intact if pictures were shown of them after the isolation, preferably after sorting. Is the sorting system capable of dealing with cells that are 1mm long? The RNA content of spermatids is reported to be low in this data set. It could be, but it could also be that these cells are not intact in the sample, and only a fragment of the cell has made it into the sequencing. Clearly recovery of late spermatids has been inefficient in this sequencing as there are only 84 such cells from the 5000 cells sequenced. About half of the germline cells in the testis are post-meiotic, so a much higher number of spermatids is to be expected. Similarly, what happens to the cyst cells after they have been dissociated? Their normal morphology is very flat (and concave, wrapped around the germline). The results suggest that these cells express many genes, but again "cells" in this class could also include fragments of other cell types – the relatively high expression of *fzo, twe, soti, Dpy-30L2* and *p-cup*, all genes that are reported to be germline specific in this class is therefore of concern.

6) The integrity of the starting cell population probably does not majorly impact the most abundant cell types spermatogonia and early vs late spermatocytes, however I am not sure the transitions between these cell types are accurately annotated. – "We found that *twe* expression peaked later in spermatogenesis (546) than *fzo*, and concluded that clusters expressing *twe* but not *fzo* were late spermatids." No, both these transcripts are present in early-mid and late primary spermatocytes, indeed *fzo* is not translated until the cells are secondary spermatocytes. *twe* translation is somewhat earlier. These two genes have very similar expression patterns according to RNA in situ. The data for *twe* and *fzo* in particular in Figure 3 do not show a strong distinction. Most of the cells in the dataset have reasonable levels of both genes suggesting that most cells are really primary spermatocytes. It would be better to use a gene that is transcribed earlier in spermatocytes, and declines as spermatocytes mature, to distinguish these cell types – eg *aly* or rye (Taf12L, PC2 contributor from Figure 1—figure supplement 1), or the ribosomal proteins.

7) We do not dispute the pseudotime trajectory that is calculated, it is the boundaries between classes that we are not sure of. In Figure 3 the transition between state 1 and state 2 seems to be at the very end of differentiation, forced on the timeline by the artefactual branch of somatic cells. There are surely more real biological states in this series, and the actual boundaries could be more accurately mapped. E.g., the rapid decline in the number of transcripts present per cell at about 60% of pseudotime likely reflects the transition from spermatocytes through meiosis to early spermatid elongation – a phase at which many transcripts are degraded.

8) The class named as mature spermatids are probably not fully mature.

9) Figure 4 would be much easier to interpret if it also included z-score tracks on the pseudotime plot for the markers used in assigning cell type for reference (i.e., *bam, vasa* etc).

10) Figure 4—figure supplement 1 is not possible to interpret as shown. The order of genes appears to be different for the parental vs child sets. What this figure needs to show is each pair of genes and their expression in pseudotime, aligned with each other, not clustered within their own class.

11) Abstract. "the distribution of new genes across cell types" is a misleading phrasing. All cell types have all genes. We are not sure exactly what is meant here. Introduction " The testis is a highly heterogeneous tissue", is it really more heterogenous than other tissues (gut, brain, etc?).

12) Results paragraph five and the following: please explain on what basis these de novo genes were identified. Were they identified from transcript in the male gonads? If yes, they are not a random set of de novo genes and may be biased depending on how the mRNA data was collected in the original study. It would be more useful to identify de novo genes from the data itself and look at their age, population frequency, stage at which they are seen, etc. If the original paper just looked at bulk testis expression, the most abundant cell types in the extract will be overrepresented, biasing the discovery of de novo genes in the stages with more coverage. Since the main result of the paper relates to de novo genes, this part of the analysis needs to be strengthened.

13) Results paragraph six. The authors show that de novo genes are initially expressed in some specific cell types and they conclude that spermatocyte expression is an important step for de novo origination. Important is vague term and the reader does not understand what is being shown and supported.

14) Figure 2C: The authors write: "For every cell type except spermatocytes, segregating de novo genes are the least commonly expressed". It seems that segregating de novo genes are always the least expressed, including in the spermatocytes. Also, this analysis shows that in early spermatocytes, fixed de novo genes are more expressed that all other genes. This is an example of bias that I was mentioning in my second point. If this stage is more represented in the study that identified de novo genes, it is not surprising that the de novo genes studied here are more expressed there because they needed to be highly expressed to be discovered in the first place.

15) The comparison with gene duplicates needs more work. The authors need to make sure the duplicates are matched with the de novo genes in terms of their age and maybe also size since detectability by RNAseq could be biased towards longer transcripts. Finally, the mechanisms of duplication needs to be taken into account (RNA based gene duplication versus tandem duplication versus other mechanisms). Very young duplicates are also probably difficult to differentiate at the RNA level.

16) The high proportion of fixed de novo genes expressed in spermatocytes is not unexpected, and fits with the theory that spermatocyte chromatin is permissive for transcription. Most of these genes are testis-specifically expressed, and most testis-specifically expressed genes are expressed in spermatocytes. The "all genes" class includes genes that are expressed earlier in the germline and in somatic cells. If testis-specifically expressed old genes were analysed they would likely have the same pattern as the de novo genes.

---

## [Author Response]

Essential revisions:1) For non-drosophilist readers, it is very important that the authors explain early on that male Drosophila do not typically undergo meiotic recombination. This has important implications for the interpretation of some of the results (especially the section on mutational load).

Thank you for bringing up this point. A sentence clarifying this point has been added to the Introduction, and Discussion subsection “Gene age and mode of origination affects gene expression bias across cell types”.

2) The single cell sequencing experiments were conducted on a pool of testes from 50 flies. We can appreciate that single cells are being profiled and this is some kind of within experiment replication, but this does not provide us with information on how reproducible the entire experiment is. The authors may have access to more data or independent data that would allow to demonstrate the reproducibility of the analyses. If it is the case, it would be useful to provide an analysis of such data.

Two analyses demonstrating the high degree of reproducibility of our experiments were added in the new version of the manuscript. First, we checked the expression correlation between a bulk RAL517 RNA-seq data (Zhao et al., 2014) and the “bulked” RAL517 single-cell RNA-seq data (treating single-cell RNA seq data as bulk data), these data correlate with a Pearson’s R of 0.97, indicating that the single-cell sequencing data has accurately captured the testis transcriptome in a reproducible fashion. This suggest that although there may be bias for single-cell sorting, this bias is very small and may largely be biased against late spermatid and mature sperm. In addition, we correlated TPM’s from our sample with a separate library from a different *D. melanogaster* strain from our lab, also with a Pearson’s R of 0.97. As such, our data are robust with respect to biological and technical variation. This also suggest that although there might be some degree of batch effects for single-cell RNA-seq data, this is unlikely impact RNA expression pattern globally. This result was added to Figure 1—figure supplement 2.

In addition, it is unclear to me how single-cell RNA-seq data is normalized (or not) and what the variance level is among cells. This seems like an important point that needs to be considered (whether the authors think this is a problem or not).

To normalize our data, we used Seurat’s normalizeData function with default parameters. Regarding normalization, the program used log-normalization. Specifically, according to Seurat’s documentation, “Feature counts for each cell are divided by the total counts for that cell and multiplied by the scale.factor (default=10,000). This is then natural-log transformed using log1p.” This log-normalization and a subsequent scaling step ensure that higher transcriptional activity of a given cell type does not bias our ability to infer relative gene expression across cell types. We expanded the Materials and method.

If spermatocytes have overall higher transcriptional activity than other cell types (which could be the case considering more genes and mRNA molecular are detected, Figure 2A,B), this could in turn increase power to detect de novo genes, especially if these tend to be expressed at low levels (and affect some conclusions drawn, e.g. de novo genes to be expressed in higher proportion at the early spermatocyte stage, Figure 2C).

We acknowledge that our study only seeks to analyze previously characterized de novo genes, and will inherit the limitations of Zhao et al., 2014, the source paper for these segregating and fixed de novo genes. Zhao et al., 2014 detected de novo genes from bulk testis RNA-seq of multiple *D. melanogaster* strains, meaning that de novo genes that are enriched in a rare cell type may not be counted as de novo genes if their expression in the whole tissue does not reach a certain threshold. Despite this possibility we still observe many de novo genes with maximum expression in rare cell types such as germ line stem cells and spermatogonia. This shows that we are able to detect de novo genes from less abundant/transcriptionally active cells. In future studies, when the cost of single-cell sequencing comes down, we hope to use it to identify de novo genes that may be more cell-type specific than the genes currently known. These points are addressed in subsection “Preparation of custom annotation file for de novo gene analysis”.

3) The identification of de novo mutations needs to be supported or discussed. Two reviewers were convinced but another one raised these points. It is surprising that many mutations are identified. Also, the analysis needs to be refined. The tissues examined are from a pool of flies. Are mutations from different individuals? 73 substitutions means about 1.5 substitution per fly since 50 flies are studied. Are mutations identified at one stage carried on to the next stage in the data? The per-base mutation rate would need to take into account the number of cell divisions, no? Are all stages going through cell division?

Putative de novo mutations are each likely unique to an individual. If a mutation were found in multiple individuals, it would likely be an inherited somatic variant and we would catch such variant alleles in somatic cells. For each of the mutations, we identified reads from somatic cells with the WT allele at that position, and the mutated allele is only present in germ cells. Each variant is also supported by multiple germ cell reads with different UMIs.

Since the data is a snapshot of every cell in the testis at the same time, we would expect de novo variants to be present in a single cell type since cells from a common progenitor differentiate simultaneously (except one daughter cell from GSC division). In reality, we observe some variants present in adjacent cell types, reflective of the fact that transitions between these cell types are a continuous, not binary phenomenon.

We agree with the reviewer that “73 substitutions mean about 1.5 substitution per fly since 50 flies are studied”. However, because 73 substitutions are unlikely to reflect the total mutations in all 50 flies given that we only sequenced a small subset of the cells from the 50 flies, we think it would not be the most meaningful way to report the data. Additionally, some of the 77 positional variants are linked, leaving 44 total mutational events, not 73 (this is already accounted for in our previous analysis). We are hesitant to include the number of cell divisions in our calculated mutational load because based on the current resolution it is impossible to know during which cell division a variant arose, and some of our assigned cell types encompass multiple cell division cycles. We do agree with the reviewer that it is an interesting and important question to pursue in the future when the technique is advanced enough to provide such resolution. We have updated the manuscript to emphasize these points in subsection “Mutational load decreases throughout spermatogenesis” paragraph two.

4) Could the authors make a prediction as to what active lesion repair vs death of cells carrying unrepaired lesions would look like in their data? I think the latter is more likely than the former given the observations (and that this is a very important and exciting result that could stand out more), because I predict that lesions would be difficult to transcribe (and perhaps lead to various nt misincorporation?), but I could be wrong. If correct, this result has important implications with respect to selective pressure within the testis.

It is a very exciting and important question to consider, and we also hypothesize that the second model is likely. However, although we tried to identify differentially expressed cell-death related genes in the mutated cells, we were not able to distinguish these two models based on current knowledge. One of the major challenges is that there are not enough marker genes (especially cell-death related genes in *Drosophila* testis) to be used to provide a statistical power. We are not aware of other ways to distinguish between active lesion repair and death of cells carrying unrepaired lesions, since either will result in removal of mutated cells. In the future, we or other colleagues in the field might be able to do a lineage-tracing experiment to show that cells transcribing a certain transgenic marker acquire and lose mutations during spermatogenesis. To make this point clearer, we expanded it in the Discussion subsection “Gene age and mode of origination affects gene expression bias across cell types”.

5) The critical point of this paper in terms of determining whether the conclusions are supported by the data is the generation and interpretation of the single cell sequencing data, and one expert reviewer remains to be convinced by all of it. She would be much more reassured that the cells are healthy and intact if pictures were shown of them after the isolation, preferably after sorting.

High-quality data is the most important first step for this work. We appreciate the caution and the detailed point from the reviewer. In the original submission, we reported that the single cells with an average of 93-96% viability after sorting, this is a very high number of live cells for single-cell suspension. The other WT strain suspension yielded a similar viability of ~95%. To address her concern, we have included pictures of the single-cell suspension of the sequenced strains created with our protocol showing that cells are intact and form a range of sizes (Figure 1—figure supplement 1). A very important point about the figures is that cells in vitro do not have the same observed size as in vivo, because cells are rarely perfectly round in vivo. Without physical forces from nearby cells, the vast majority of the cells in vitro are round cells.

Is the sorting system capable of dealing with cells that are 1mm long? The RNA content of spermatids is reported to be low in this data set. It could be, but it could also be that these cells are not intact in the sample, and only a fragment of the cell has made it into the sequencing. Clearly recovery of late spermatids has been inefficient in this sequencing as there are only 84 such cells from the 5000 cells sequenced. About half of the germline cells in the testis are post-meiotic, so a much higher number of spermatids is to be expected. Similarly, what happens to the cyst cells after they have been dissociated? Their normal morphology is very flat (and concave, wrapped around the germline). The results suggest that these cells express many genes, but again "cells" in this class could also include fragments of other cell types – the relatively high expression of fzo, twe, soti, Dpy-30L2 and pcup, all genes that are reported to be germline specific in this class is therefore of concern.

As mentioned in the above paragraph, we see variation in cell size – though the cell size differences are not as dramatic as in vivo – but not shape, and see no clusters of adhered cells or intact cysts. We did not observe evidence of fragmented cells or cell debris. It seems clear that the cytoplasmic bridges between conjoined germ cells have been broken and these have been separated from cyst cells. The trypsin, collagenase and filtration steps appear to have coerced these cell types present in our sample into a spherical shape, and none appeared larger than 30 uM, the maximum size allowed by the chromium controller. We saw spermatid-like cells with identifiable tails, and in our sequencing data we see clear evidence of their presence- the cluster-specific enrichment of *p-cup* and other late-spermatid genes. Their long shape means that spermatids must go through the Chromium controller channel in a precise orientation to be captured, resulting in a lower capture efficiency than other cells. However, although *Drosophila* spermatids have long tails, they are not necessarily linear in solution, and the tail sometimes wraps around the cell body in a ball shape outside the testis. Although it is likely that the capture of these late-stage cells is inefficient, the captured cells show reasonable expression profiles of spermatid-specific genes. We agree with the reviewer that the testis contains many more spermatids than we have captured. In combination with response (2) that the single-cell sequencing result is highly consistent with bulk RNA-seq data, these results suggest that while some degree of bias is likely, the impact of this bias is relatively small.

Related to the shape of cyst cells, because the cells are not attached to each other, they assume a round or near round shape within the cell suspension. Regarding the expression of *fzo, twe, soti, DPY-30L2* and *p-cup* in cyst cells, we agree that expression of these genes is unexpected, which was largely due to the data presentation of Figure 1D. The clusters marked as cyst cells show specific enrichment of zfh1, indicating that this cluster definitely contains cyst cells. The original scaled color may be misleading, and we adjusted the scaled expression color pattern in Figure 1D to better show cluster-specific enrichment of key marker genes, and found that cyst cells show that *fzo* and *twe* were somewhat expressed in this cluster, but not *Dpy-30L2* or *p-cup* compared to other non-spermatid clusters, consistent with Figure 1C. This indicates that this cluster could contain cyst cells as well as some transcriptionally similar or physically adhered spermatocytes.

Our assigned spermatocyte clusters show no enrichment of *Fas3*, indicating that we have been conservative in our assignment of germ cells at the expense of some false negatives. While our assigned germ cell clusters appear to be relatively free of somatic cells, the somatic cell clusters may contain some misidentified germ cells. Our difficulties identifying true cyst cells indicate that for single-cell analysis of the cyst cell lineage, a future study might need to perform a FACS-purification step to remove germ cells or doublets.

6) The integrity of the starting cell population probably does not majorly impact the most abundant cell types spermatogonia and early vs late spermatocytes, however I am not sure the transitions between these cell types are accurately annotated. – "We found that twe expression peaked later in spermatogenesis (546) than fzo, and concluded that clusters expressing twe but not fzo were late spermatids." No, both these transcripts are present in early-mid and late primary spermatocytes, indeed fzo is not translated until the cells are secondary spermatocytes. twe translation is somewhat earlier. These two genes have very similar expression patterns according to RNA in situ. The data for twe and fzo in particular in Figure 3 do not show a strong distinction. Most of the cells in the dataset have reasonable levels of both genes suggesting that most cells are really primary spermatocytes. It would be better to use a gene that is transcribed earlier in spermatocytes, and declines as spermatocytes mature, to distinguish these cell types – eg aly or rye (Taf12L, PC2 contributor from Figure 1—figure supplement 1), or the ribosomal proteins.

Thank you for this comment. We apologize for a typo- “clusters expressing *twe* but not *fzo* were late spermatids” should have read “late spermatocytes.” We have since corrected it, and given that our observations about the peak expression patterns of *fzo* and *twe* have not been previously reported, we have sought to strengthen our early/late spermatocyte assignments with another gene, *aly*, which is known to decline throughout spermatogenesis. We have added *aly* to Figure 1D. As predicted by the reviewer, it is less commonly expressed in our assigned late spermatocytes than in early spermatocytes, strengthening our confidence in our cell-type predictions.

7) We do not dispute the pseudotime trajectory that is calculated, it is the boundaries between classes that we are not sure of. In Figure 3 the transition between state 1 and state 2 seems to be at the very end of differentiation, forced on the timeline by the artefactual branch of somatic cells. There are surely more real biological states in this series, and the actual boundaries could be more accurately mapped. E.g., the rapid decline in the number of transcripts present per cell at about 60% of pseudotime likely reflects the transition from spermatocytes through meiosis to early spermatid elongation – a phase at which many transcripts are degraded.

We completely agree with this comment and suggestion that state 2 is not a real biological state. However, assigning discrete biological states along a pseudotime trajectory is tricky since the cell states exist not in discrete blocks, but in a continuum of gene expression. The most real, discrete biological state in our pseudotime trajectory is the difference between somatic cells and germ cells, so we have recolored Figure 3A to reflect this.

8) The class named as mature spermatids are probably not fully mature.

We revised this term, and these cells are now called “late spermatids”.

9) Figure 4 would be much easier to interpret if it also included z-score tracks on the pseudotime plot for the markers used in assigning cell type for reference (i.e., bam, vasa etc).

This is a great suggestion and we have added a marker track to Figure 4 as a visual aid.

10) Figure 4—figure supplement 1 is not possible to interpret as shown. The order of genes appears to be different for the parental vs child sets. What this figure needs to show is each pair of genes and their expression in pseudotime, aligned with each other, not clustered within their own class.

Figure 4—figure supplement 1 now includes all *D. melanogaster*-specific duplicate genes detected in our library, aligned with their parental copies.

11) Abstract. "the distribution of new genes across cell types" is a misleading phrasing. All cell types have all genes. We are not sure exactly what is meant here. Introduction " The testis is a highly heterogeneous tissue", is it really more heterogenous than other tissues (gut, brain, etc?).

We revised the sentences to “To investigate the expression patterns of genetic novelties across different testis cell types, we performed single-cell RNA-sequencing of adult *Drosophila* testis. We found that new genes were expressed in various cell types, the patterns of which may be influenced by their mode of origination” and “The testis is a highly transcriptionally active tissue”.

12) Results paragraph five and the following: please explain on what basis these de novo genes were identified. Were they identified from transcript in the male gonads? If yes, they are not a random set of de novo genes and may be biased depending on how the mRNA data was collected in the original study. It would be more useful to identify de novo genes from the data itself and look at their age, population frequency, stage at which they are seen, etc. If the original paper just looked at bulk testis expression, the most abundant cell types in the extract will be overrepresented, biasing the discovery of de novo genes in the stages with more coverage. Since the main result of the paper relates to de novo genes, this part of the analysis needs to be strengthened.

In Zhao et al., 2014, segregating and fixed de novo genes were identified based on transcription in the testis of several strains of *D. melanogaster*. They were characterized as segregating or fixed based on their frequency across different strains. Since our single cell preparation correlates extremely well with whole-tissue RNA sequencing (Figure 1—figure supplement 2), it is reasonable to examine previously identified de novo genes from whole tissue. Identifying segregating de novo genes from a single-cell preparation would be ideal, but is not possible now because it requires libraries from multiple *D. melanogaster* strains and libraries from outgroup species with ultra-deep sequencing that is beyond the scope of this study. Until the costs of single-cell sequencing come down, using previously identified de novo genes from bulk testis RNA-seq data is the best available option. We do completely agree with the reviewers that genes that are expressed in the less abundant cell types are difficult to identify and study, which is a challenge using the RNA-seq method. We discussed this point in the Materials and methods.

13) Results paragraph six. The authors show that de novo genes are initially expressed in some specific cell types and they conclude that spermatocyte expression is an important step for de novo origination. Important is vague term and the reader does not understand what is being shown and supported.

We toned down the statement. Given our new finding that early spermatocytes express a greater fraction of fixed de novo genes than testis-specific genes or all other genes, this section now reads: “The high proportion of de novo genes expressed in spermatocytes suggests that such genes may play functional roles in these cells and development stage.”

14) Figure 2C: The authors write: "For every cell type except spermatocytes, segregating de novo genes are the least commonly expressed". It seems that segregating de novo genes are always the least expressed, including in the spermatocytes. Also, this analysis shows that in early spermatocytes, fixed de novo genes are more expressed that all other genes. This is an example of bias that I was mentioning in my second point. If this stage is more represented in the study that identified de novo genes, it is not surprising that the de novo genes studied here are more expressed there because they needed to be highly expressed to be discovered in the first place.

We apologize for the misunderstanding. Figure 2C shows that a lower proportion of segregating de novo genes are detected in a given cell type or cluster, but this does not indicate that they were lowly expressed in a given cell type. We added a sentence reading “It is important to note that this measure looks at the number of genes of each type detected, not the expression level of each, and does not distinguish between high and low expression”. Related to the last point that we may have more power to detect gene expression and expression bias for enriched cell types, we overall agree with the reviewer’s point, which was discussed and addressed in response 12.

15) The comparison with gene duplicates needs more work. The authors need to make sure the duplicates are matched with the de novo genes in terms of their age and maybe also size since detectability by RNAseq could be biased towards longer transcripts. Finally, the mechanisms of duplication needs to be taken into account (RNA based gene duplication versus tandem duplication versus other mechanisms). Very young duplicates are also probably difficult to differentiate at the RNA level.

This is a great point and we have revised Figure 4—figure supplement 1, and accompanying text in the revised version. We agree that it is important to compare young duplicate genes with de novo genes of the same age, and have accordingly corrected Figure 4 to show only *melanogaster*-specific de novo and duplicate genes. We can only detect the expression of 14 of these duplicate genes, however, and 12 of them are tandem duplicate genes, and 2 are retroposed genes. This means that while we can make a fair comparison of expression patterns between *melanogaster*-specific duplicate genes and de novo genes, just using *melanogaster*-specific genes we cannot make inferences on the relationship between duplication mechanism and testis expression pattern.

If child duplicate genes are indistinguishable from their parental copies at the sequence level, the two copies would show the same general expression pattern across spermatogenesis. We only observed this for a small number of parental/child gene duplicates, giving us confidence that this is not a significant source of bias in our data (Figure 4—figure supplement 1).

16) The high proportion of fixed de novo genes expressed in spermatocytes is not unexpected, and fits with the theory that spermatocyte chromatin is permissive for transcription. Most of these genes are testis-specifically expressed, and most testis-specifically expressed genes are expressed in spermatocytes. The "all genes" class includes genes that are expressed earlier in the germline and in somatic cells. If testis-specifically expressed old genes were analysed they would likely have the same pattern as the de novo genes.

This is a keen observation, and we have therefore revised the analysis to account for such bias by adding testis-specific genes to our scaled expression comparisons in Figures 2, 4, and 5. We defined testis-specific genes as any gene that, in FlyAtlas2 data, had FPKM >2 in testis and <1 in every other tissue (except whole male). In Figure 4A we can now see that in most cell types de novo genes tend to show scaled expression between all other genes and testis-specific genes. In late spermatogonia and early spermatocytes, fixed de novo genes are expressed similarly to testis-specific genes, while segregating de novo genes are rarer.